# Next Generation Sequencing in AML—On the Way to Becoming a New Standard for Treatment Initiation and/or Modulation?

**DOI:** 10.3390/cancers11020252

**Published:** 2019-02-21

**Authors:** Michael Leisch, Bettina Jansko, Nadja Zaborsky, Richard Greil, Lisa Pleyer

**Affiliations:** 1Department of Internal Medicine III with Haematology, Medical Oncology, Haemostaseology, Infectiology and Rheumatology, Oncologic Center, Salzburg Cancer Research Institute-Laboratory of Immunological and Molecular Cancer Research (SCRI-LIMCR), 5020 Salzburg, Austria; m.leisch@salk.at (M.L.); b.jansko@salk.at (B.J.); n.zaborsky@salk.at (N.Z.); r.greil@salk.at (R.G.); 2Paracelsus Medical University, 5020 Salzburg, Austria; 3Cancer Cluster Salzburg, 5020 Salzburg, Austria

**Keywords:** AML, acute myeloid leukemia, next generation sequencing, NGS, targeted therapy, minimal residual disease

## Abstract

Acute myeloid leukemia (AML) is a clonal disease caused by genetic abberations occurring predominantly in the elderly. Next generation sequencing (NGS) analysis has led to a deeper genetic understanding of the pathogenesis and the role of recently discovered genetic precursor lesions (clonal hematopoiesis of indeterminate/oncogenic potential (CHIP/CHOP)) in the evolution of AML. These advances are reflected by the inclusion of certain mutations in the updated World Health Organization (WHO) 2016 classification and current treatment guidelines by the European Leukemia Net (ELN) and National Comprehensive Cancer Network (NCCN) and results of mutational testing are already influencing the choice and timing of (targeted) treatment. Genetic profiling and stratification of patients into molecularly defined subgroups are expected to gain ever more weight in daily clinical practice. Our aim is to provide a concise summary of current evidence regarding the relevance of NGS for the diagnosis, risk stratification, treatment planning and response assessment in AML, including minimal residual disease (MRD) guided approaches. We also summarize recently approved drugs targeting genetically defined patient populations with risk adapted- and individualized treatment strategies.

## 1. Introduction

Acute myeloid leukemia (AML) is an aggressive hematologic malignancy characterized by uncontrolled proliferation and accumulation of immature myeloid precursor cells in the bone marrow leading to impaired hematopoiesis and bone marrow failure [1]. It is the most common acute leukemia in adults with an incidence of up to 17 per 100,000 per year in patients older than 65 years [2].

AML is a clonal disease caused by genetic mutations in normal myeloid hematopoetic progenitor cells, leading to altered self-renewal, differentiation and proliferation [1]. In the last decade, there has been a tremendous increase in our knowledge regarding the mutational landscape of AML [1,2,3], largely based on advances in sequencing techniques. Next generation sequencing (NGS) is a massively parallel sequencing technology that allows for rapid, precise and cost-effective sequencing of multiple genes or even whole exomes and genomes within a single day, and has revolutionized genomic research. This has led to changes in the classification, prognostic stratification, treatment and response assessment of AML.

Here, we systematically review current literature on the use of NGS for the diagnosis, risk stratification, treatment initiation and/or modulation, as well as for response assessment in AML. We conducted a literature search using the PubMed database and screened titles and abstracts with relevant data for the use of NGS in AML (search terms used: AML or acute myeloid leukemia and NGS or next generation sequencing). For targeted therapies we used the search term AML or acute myeloid leukemia and filtered the results for clinical trials within the last 10 years. Papers were included if NGS was used for either diagnostic, prognostic or therapeutic purposes or for response assessment. Medical therapy was included if a molecularly targeted therapy was used alone or in combination with other substances. We searched clinicaltrials.gov for ongoing trials with targeted agents.

The aim of the treatment part of this review is not to give an overview of AML treatment per se, but to specifically summarize which substances are thought to specifically target certain mutations (or groups of mutations), and thus might be used as a result of mutational profiling by NGS (or other conventional means such as RT-PCR).

## 2. Next Generation Sequencing

### 2.1. Terminology

Next-generation sequencing (NGS), also known as high-throughput sequencing, is the term used to describe a number of different modern massively parallel sequencing technologies. For clarification of termini, ‘(ultra) deep sequencing’ refers to NGS approaches enabling the detection of rare clonal cells comprising 1% or less of the original sample (https://emea.illumina.com/science/technology/next-generation-sequencing/deep-sequencing.html?langsel=/at/; accessed on 4 January 2019).

### 2.2. Transition of NGS from Research to Clinical Practice and Market Overview

NGS used to be an extremely expensive and time-consuming procedure. The costs for sequencing a whole human genome have dramatically sunk in the last two decades: from a phenomenal 100 million USD in 2001, to approximately 1000 USD in 2015; similarly, the costs for sequencing per raw megabase sunk from approximately 10.000 USD in 2001 to less than 10 cents in 2011 (https://www.genome.gov/27565109/the-cost-of-sequencing-a-human-genome/).

Illumina (San Diego, CA, USA) platforms use sequencing by synthesis methodologies. The iSeq100, miniSeq, miSeq and the nextSeq System are Illuminas benchtop sequencers. They also have the high-throughput platforms HiSeq2500, HiSeqX Ten and the latest NovaSeq.

Thermo Fisher Scientific (Waltham, MA, USA) offers the Ion Proton System, Ion PGM System, Ion S5 System, Ion S5 XL System, Ion GeneStudio S5 System and the HID GeneStudio S5 System. Using semiconductors, these platforms work via measuring the pH changes resulting from hydrogen ions released during the addition of nucleic acids. Pacific Bioscience (Menlo Park, CA, USA) offers the PacBio System platform, using single molecule realtime technology. Oxford Nanopore Technology (Oxford Science Park, Oxford, UK) have electronics-based DNA/RNA sequencing platforms. They use nanopore sequencing, which measures changes in the electrical current when molecules pass the nanopore. They offer a portable pocketsize MinION sequencing device and two benchtop sequencers: GridION and PromethION, which can run MinION flowcells. Their smallest device so far is the SmidgION for portable DNA analysis, which can even be used with a smartphone.

There are huge differences in costs between whole genome sequencing (WGS), whole exome sequencing (WES) and targeted sequencing (where chosen regions of interest are sequenced). Depending on sample number, read size, number of megabases in the panel, the quality control methods applied and the sequencing platform used, the costs for targeted sequencing range between ~200–1000 € or more per run, and the running time of the sequencing analyses may range anywhere from 2 h to a few days.

Currently, commercially available myleoid NGS gene panels include: (i) SureSeq myPanel™ NGS Custom AML (Oxford Gene Technology, Begbroke, Oxfordshire, UK); (ii) Leuko-Vantage Myeloid Neoplasm Mutation Panel (Quest Diagnostics, Madison, NJ, USA); (iii) AmpliSeq^®^ Myeloid Sequencing Panel (Illumina); and (iv) Human Myeloid Neoplasms Panel (Qiagen, Venlo, The Netherlands). A comparison of the covered genes is provided in Table 1.

## 3. NGS for the Diagnosis of AML

### 3.1. AML Classification and Pathogenesis

The diagnosis of AML requires the presence of at least 20 percent blast forms of myeloid origin in a bone marrow aspirate or peripheral blood smear [4]. Irrespective of blast cell count, AML can also be diagnosed when certain genetic abnormalities are present (i.e., t(8;21), inv(16), t(15;17)) or in the case of myeloid sarcoma [4]. AML is currently classified according to the 2016 revision to the WHO classification of myeloid neoplasms and acute leukemia [4]. Herein, AML is further subclassified into: (i) AML with recurrent genetic abnormalities (including t(8;21); inv(16); PML-RARA; t(9;11); t(6;9); inv(3); t(1;22); BCR-ABL; nucleophosmin (NPM1) mutations; biallelic CEBPA mutations and AML with RUNX1 mutations as a provisional entity), (ii) AML with myelodysplasia related changes, (iii) therapy related myeloid neoplasm, (iv) AML, not otherwise specified, (v) myeloid sarcoma and (vi) myeloid proliferations related to Down Syndrome.

In the 2016 update of the WHO classification for AML with recurrent genetic abnormalities, AML with NPM1 and biallelic CEBPA mutations became full entities with a favorable prognosis. In addition, AML with RUNX1 mutations was added as a provisional entity, as patients with RUNX1 mutations often present with immature morphology and/or secondary AML with an inferior prognosis [5].

Several large sequencing studies conducted in the last decade revealed the genetic heterogeneity of the disease [1,2,3] (Table 2) and gave further insights into the molecular pathogenesis of AML. In 2013, The Cancer Genome Atlas (TCGA) network reported on the mutation frequency of 200 AML- and matched normal skin samples analysed by either WGS (*n* = 50) or WES (*n* = 150) [2]. On average, 13 mutations were detected per patient and mutations in 23 genes were found to be recurrently mutated. Mutations in another 237 genes were detected only in a minority of patients [2]. The 23 recurrently mutated genes were grouped into nine functional categories (i.e. transcription-factor fusions (18% of cases), NPM1 mutations (27%), tumorsuppressor genes (16%), DNA-methylation-related genes (44%), activated signaling genes (59%), chromatin-modifying genes (30%), myeloid transcription-factor genes (22%), cohesin-complex genes (13%), and spliceosome-complex genes (14%) [2]. Furthermore, this study also analyzed the clonal structure of AML according to the variant allelic frequency (VAF) of the detected mutations: about 50% of the patients had at least one subclone in addition to the founding clone.

The results discussed above led to the new concept of premalignant stages preceeding the evolution of AML as part of a multistep clonal evolutionary process. Initiating mutations at low variant allelic frequency (VAF), so called “passenger lesions” (including TET2, DNMT3A, GNAS, ASXL1, SF3B1, PPM1D) usually occur early in the disease course and are not sufficient to cause AML per se [2,6,7,8,9,10]. These mutations are also found in blood samples of a substantial proportion (10–20%) of healthy, usually older, individuals without a hematologic disease [6,7,8,9,10]. This state has been termed ‘clonal hematopoiesis of indeterminate potential’ (CHIP). Individuals with CHIP harbor a higher risk for development of a myeloid neoplasm (about 1% per year), but also have for example a two fold higher risk for the development coronary heart disease [11]. The term clonal hematopoiesis with substantial oncogenic potential (CHOP) is used by some authors for individuals who are found to have a higher risk, disease defining mutation (so called “driver mutations” like BCR-ABL, JAK2, Runx, FLT3, KRAS, HRAS at higher VAF) without fulfilling other diagnostic criteria of the disease of as reviewed recently [12,13]. 

With the acquisition of further mutations, the clinically overt AML phenotype subsequently develops over time. Similar pre-leukemic conditions with a higher risk for the development of e.g., MDS have been recently recognized and include: (i) idiopathic cytopenia of undetermined significance (ICUS), (ii) idiopathic dysplasia of undetermined significance (IDUS) and (iii) clonal cytopenia of undetermined significance (CCUS), [13] in analogy of monoclonal gammopathy of undeterminate potential (MGUS) as a precursor for multiple myeloma [23]. Figure 1 gives an overview of this evolutionary process.

### 3.2. Current Guidelines Regarding NGS Analyses for the Diagnosis of AML

Currently, the European Leukemia Network (ELN) recommends genetic testing for all patients with newly diagnosed AML [24]. This includes: (i) conventional cytogenetics, (ii) screening for (a minimum of) six gene mutations including NPM1, CEBPA, RUNX1, FLT3, TP53, ASXL1 and (iii) screening for gene rearrangements including PML-RARA, CBFB-MYH11, RUNX1-RUNX1T1 and BCR-ABL1 (Figure 1).

Similarily, the current National Comprehensive Cancer Network (NCCN) guidelines recommend conventional cytogenetics and testing for gene rearrangements identically to the ELN recommendations. The ELN recomends testing for (a minimum of) 9 gene mutations including NPM1, CEBPA, RUNX1, FLT3, TP53, ASXL1, IDH1, IDH2 and c-KIT (www.nccn.org, accessed on 12 November 2018) (Figure 1).

Both institutions acknowledge that the recommended mutational testing has to be interpreted as a “minimum” for daily clinical practice in order to accurately assess genomic risk and use targeted therapy where appropriate. According to the NCCN and ELN, multiplex gene panels and NGS analysis may be used to obtain further information regarding prognosis, treatment decisions and eligibility for clinical trial participation [24].

Taken together, the genomic and molecular classification of AML is still a field of active research and further changes in the classification system are likely to evolve in the future. In our opinion, a NGS panel for clinical use has to cover mutations that are of diagnostic, prognostic (i.e. inform about the patient’s prognosis) and predictive (i.e. predict response to a certain (targeted) therapy) relevance. Currently there is no consensus regarding the number or composition of genes to be inbcluded in a myeloid NGS panel. Table 2 summarizes the genes analysed, and the mutation frequency reported by all major publications in the field of AML using NGS technologies and Table 1 summarizes currently commercially available NGS panels.

## 4. Prognostic Implication of Gene Mutations in AML

### 4.1. Overview of AML Prognosis and Age

Prognosis in AML is dependent on patient related factors (mainly age and comorbidities) and disease related factors. Age per se, and the accompanying decrease in organ functions, is one adverse prognostic risk factor, however, older patients (>65 years) also tend to have a lower incidence of favourable risk karyotype and a higher amount of adverse cytogenetic features, as compared to younger adults [25,26]. Older patients (>65 years) also show a trend towards higher numbers of detected mutations [27] as well as a higher incidence of secondary AML with adverse genetic mutations (i.e., SF3B1, SRSF2) and inferior outcomes compared to younger patients [19]. Disease related factors are largely reflected by the genetic composition of AML as discussed below.

### 4.2. Conventional Cytogenetics

At the time of diagnosis, chromosomal abberations can be detected in about half of the patients with conventional cytogenetics [28]. Preatreatment karyotype has been the earliest genetic prognostic scoring system. Several large prospective trials evaluated patient outcomes after intensive chemotherapy based on pretreatment karyotype including: Medical Research Council (MRC) [28], Cancer and Leukemia Group B (CALGB) [29,30], Southwest Oncology Group (SWOG) [31], French AML Intergroup [32], German AML Intergroup [33], MD-Anderson classification [34]. For patients with AML treated unintensively with azacytidine, a prognostic scoring system based on performance score, white blood cell count and pretreatment cytogenetics (the European ALMA score) was developed and validated in 2015 [35].

### 4.3. Conventional Cytogenetics Refined with Analysis of Single Gene Mutations

As mentioned above, the genetic landscape of AML is highly heterogenous [1,2,3]. There have been several attempts to further improve the prognostic classification (especially in cytogenetically normal (CN) AML) with the use of RT-PCR or sanger sequencing for single gene mutations [36].

As an example, in 2014 Pastore et al. developed a molecular and clinical prognostic index for risk assessment in cytogenetically normal AML (called PINA) [37]. In this study, 669 AML patients from the AMLCG99 trial with CN-AML were divided into three prognostic groups (a score called PINA_OS_) according to NPM, FLT3-ITD and CEBPA mutational status. NPM1 and FLT3-ITD mutational status were assed through sequencing the exons of interest. (Exon 12 in case of NPM1 and Exon 14, 15 in case of FLT3-ITD). The CEBPA mutational status was assesd using a multiplex PCR with fragment analysis. WBC count, age and ECOG performance score_._ With these additional parameters, patients with normal karyotype could be further subclassified into low-, intermediate- and high-risk disease with a corresponding 5-year OS rate of 74%, 28% and 3% (*p* < 0.001), respectively. The score was validated in an independent cohort of 529 patients with CN-AML treated in CALGB front-line trials [37].

The currently most widely accepted genetic prognostic scoring system is the European Leukemia Net (ELN) classification from 2017 [24]. In the first edition from 2010, results from conventional cytogenetics and mutations in NPM, FLT3 and CEBPA were used to categorize patients into low, intermediate-1, intermediate-2 and high-risk disease [38]. The 2017 update now divides AML into three (instead of four) risk groups (i.e., favorable, intermediate and adverse), based on the results of conventional cytogenetics and single gene mutations in NPM 1, FLT3, biallelic CEBPA, RUNX, ASXL1 and TP53 [24]. This risk classification is currently also reflected in treatment recommendations from the NCCN, however, the NCCN classifies patients with core binding factor (CBF) AML (who have a favourable prognosis per se) and concurrent KIT mutations as intermediate risk (www.nccn.org).

### 4.4. Conventional Cytogenetics Refined with NGS Analysis of Multiple Genes

Cytogenetically defined subgroups of AML can be further refined and subclassified with NGS analysis: In 2016 Duployez et al. performed sequencing with a 40 gene panel in 215 patients with CBF AML (i.e., AML with t(8;21) or inv(16)) [39]. They found additional mutations in > 90% of patients with CBF AML. In these patients, genes involved in tyrosine kinase signaling (KIT, FLT-3 and N/KRAS) were most commonly mutated [39]. They found that mutations in epigenetic regulators (ASXL1, EZH2) and the cohesin complex were more common in AML-patients bearing t(8;21), whereas they were nearly absent in AML patients with inv(16) (42% vs. 6% for mutatiotions in epigenetic regulators, *p* < 0.001; 18% vs. 0% for cohesion complex mutations, *p* < 0.001) [39]. Mutations in ASXL1 and EZH2 were associated with a poor prognosis (HR for relapse = 5.22, *p* = 0.002) in patients with cooccuring mutations in tyrosine kinase pathways (KIT, FLT-3 and N/KRAS). Also, they found that patients with t(8;21) and a high KIT mutant allele ratio (> 35%) had an inferior prognosis compared to KIT-WT patients (5 year incidence of relapse 69.4% vs. 30.7% *p* = 0.008 for mutant vs. KIT-WT, respectively). These data suggest that diverse cooccuring mutations may influence CBF-AML pathophysiology as well as clinical behavior and point to a potential unique pathogenesis of t(8;21) and inv(16) AML, further highlighting the additional prognostic information obtainable by high throughput sequencing.

In 2016, Papaemmanuil et al. described a cohort of 1540 patients aged 18–65 years with AML treated with intensive chemotherapy [1]. Driver mutations were identified in 76 different genes in 96% of the patients [1] by a 111-fgene NGS panel (Table 2). Similar to the cancer genome atlas study [2], the authors showed that mutations in epigenetic modifiers (DNMT3A, ASXL1 and TET2) are present in early founding clones and usually coexist with other mutations, indicating that these mutations are not sufficient to cause AML and implying an evoluationary process in AML pathogenesis (Figure 1). The authors proposed a new genetic classification of AML into mutually exclusive subtypes with different biological and prognostic properties. Herein, 11 genetic subgroups of AML have been proposed, including AML with: inv(16), t(15;17), t(8;21), MLL fusion, inv(3), t(6;9), NPM1, CEBPA, TP53 aneuploidy, chromatin spliceosome and IDH2 mutations [1].

Very recently, based on the results from the study by Papaemmanuil et al., the same group later used the data set from the 1540 patients with AML to develop a knowledge bank of matched clinical and genetic data [40]. With this approach, the authors wanted to predict the benefit of allogeneic transplantation in first CR for an individual patient based on genomic and clinical variables using complex multistage statistical models. They were able to show that demographic factors (i.e., age, performance status, blood values) exerted most influence on early death rates (mostly due to treatment-related mortality), whereas genomic features most strongly influenced the dynamics of disease remission and relapse. With this approach, the authors were able to predict clinical risk and outcome more accurately than the current ELN classification. As an example, in a simulation with the developed algorithm, allogeneic HSCT could savely be omitted in 25% of the patients resulting in the same OS rate. These results have also recently been confirmed by Huet et al [41] and indicate that a refined treatment approach accounting for a patient’s individual risk may be feasible in the future when broad application of NGS will be standardized for daily clinical practice.

### 4.5. Using NGS to Predict Response to Hypomethylating Agents

Hypomethylating agents (HMAs, i.e., azacytidine and decitabine) are often used for the treatment of AML patients unfit for intensive chemotherapy [24]. Several attempts have been made to predict response to HMAs in myelodysplastic syndromes and AML. As an example, mutations in TET2 have been shown to be positively associated with response to azacytidine in some studies [42,43], however, these results where not uniformely reproduced by others [44,45,46]. On the other hand, patients with DNMT3A mutations seem to respond better to HMAs in AML [47], whereas patients with IDH mutations have been reported to have an inferior response to azacytidine [48]. Overall, single gene mutations have shown conflicting results regarding response to HMAs, therefore, pretreatment NGS analysis has also been employed.

However, in a series of 128 patients with AML and MDS treated with azacytidine, pretreatment NGS analysis with a myeloid gene panel (containing ASXL1, RUNX1, DNMT3A, IDH1, IDH2, TET2, TP53, NRAS, KRAS, EZH2, SF3B1 and SRSF2) did not reveal a prognostic significance of mutations in the tested genes. Only del 20q (assessed by conventional cytogenetics) was associated with improved survival [49].

On the other hand, Welch et al. reported a series of 86 patients with de novo or R/R AML or transfusion dependent MDS treated with a 10-day course of decitabine [22]. They performed NGS analysis of bone marrow samples at time of diagnosis and at variable time points thereafter (264 genes, Illumina HiSeq 2000 or 2500 platforms). Interestingly, they observed bone marrow blast clearance (complete remission, complete remission with incomplete count recovery, or morphologic complete remission) in 67% of patients with unfavorable karyotype versus 34% of patients with intermediate or favorable karyotype and in all patients with *TP53* mutations versus 41% of patients with wild-type *TP53* (*p* < 0.001 for both comparisons). They concluded that decitabine (in comparison to chemotherapy) resulted in favourable outcomes in this patient cohort and might mitigate the negative prognostic impact of TP53 mutations and adverse karyotype [22].

### 4.6. Using NGS to Predict Outcome after Allogeneic Transplantation

NGS has also been employed to predict outcome after allogeneic transplantation. As an example, Luskin et al. reported a series of 112 patients with AML undergoing allogeneic transplantation [50]. Using a 26 gene NGS panel (TrueSeq Custom Amplicon (TSCA), Illumina Inc.; average depth of 1500× and minimal depth of 250×, calling threshold of 5% allele frequency), 96 patients (86%) were found to have a pre-transplant mutation. They found, that mutations in TP53, WT1, and FLT3-ITD were associated with an increased risk of relapse after alloHSCT (adjusted HR (aHR) 2.90, *p* = 0.009, aHR 2.51, *p* = 0.02, and aHR 1.83, *p* = 0.07, respectively). Interestingly, six patients underwent NGS analysis prior to allogeneic transplant and at the time of relapse after alloHSCT. All six patients were found to have clonal evolution at the time of relapse. Using NGS at the time of relapse, potentially targetable mutations in FLT3-ITD, KRAS and EZH2 were found in four of six patients. Two patients were consecutively treated in experimental protocols with a FLT3 inhibitor. The authors concluded that genetic profiling is useful for assessing relapse risk in AML patients undergoing alloHSCT [50]. 

Similarly, in a cohort of 97 patients with AML (treated with intensive chemotherapy and allogeneic transplantation) and a high risk cytogenetic profile (defined as either complex karyotype, monosomy of chromosome 7, monosomy and/or deletion of the long arm of chromosome 5 and abnormalities of chromosome 17p), the presence of a TP53 mutation at diagnosis was associated with inferior outcomes after allogeneic transplantation with a three-year survival rate of 33% in patients without TP53 mutation and 10% in patients with mutated TP53 (*p* = 0.002) [51].

These results were also confirmed in another series of 113 patients with AML treated with intensive chemotherapy and allogeneic transplantation [52]. Here, Quek et al. performed sequencing of 35 genes at diagnosis and at the time of relapse (which occurred in 49 patients). They reported an increased risk of relapse in patients with pretransplant mutations in WT1 (*p* = 0.018), DNMT3A (*p* = 0.045), FLT3 ITD (*p* = 0.071), and TP53 (*p* = 0.06). Taken together, these results indicate that outcomes after allogeneic transplantation can be predicted with pre-transplant genetic profiling [52].

## 5. Next Generation Sequencing to Guide Treatment in AML?

### 5.1. Overview of Current AML Treatment Strategies

AML is an aggressive and genetically heterogenous disease. Despite this large genetic variety, most subtypes of AML have been treated uniformly over the past decades [24]. Younger (usually <55 years) and physically fit older individuals are treated with induction chemotherapy (usually combinations of cytarabine (AraC) with an anthracycline (e.g., daunorubicin or idarubicin, over a period of 7+3 or 5+2 days in dose reduced regimens, respectively) and consolidation therapy. Depending on clinical risk, consolidation for good risk AML patients (according to the ELN) consists of high dose AraC (HiDAC) chemotherapy, whereas consolidation for poor risk AML usually consists of allogeneic stem cell transplantation depending on donor availability [24]. Older and/or comorbid patients deemed unfit for intensive chemotherapy are usually treated with low dose chemotherapy or HMAs [24]. This “one size fits all” approach, does not reflect the high molecular and biological diversity of AML described above, and as such has only resulted in a median 5 year overall survival rate of 40% in patients fit for intensive chemotherapy [53]. Patients treated front-line with non-intensive strategies, such as low dose AraC (LDAC) or HMAs only have a median OS of 7–13 months [54,55,56,57,58,59,60]. Therefore, there is a significant clinical need for new, better tolerable, individualized treatment regimens, especially for older individuals.

The advances in the biological understanding of AML pathogenesis has led to the approval of eight new substances since April 2017. Considering the paucity of drug approvals for AML in the last two decades, this dramatically increases our therapeutic armentarium in the fight against AML. Recently approved drugs for AML that do not target specific mutations and hence are beyond the scope of this review, include: (i) CPX-351, the liposomal formulation of AraC and daunorubicin (FDA approved for secondary AML as front line treatment on 8 March 2017), (ii) gemtuzumab ozogamicin (FDA approved for CD33 positive AML alone or in combination with chemotherapy for newly diagnosed or R/R AML on 1 Spetember 2017), and (iii) glasdegib, an inhibitor of the hedgehog signaling pathway (FDA approved in combination with LDAC for patients with newly diagnosed AML older than 75 years or with significant comorbidities on 21 November 2018).

Recently approved drugs targeting molecularly defined patient populations include: (i) midostaurin, an inhibitor of mutated FLT3 (FDA approved for newly diagnosed AML in combination with standard chemotherapy on 28 April 2017 and on 18 Spetember 2017 by EMA), (ii) gilteritinib, and inhibitor of FLT3 and AXL (FDA approved for R/R FLT3 mutated AML on 28 November 2018), (iii) enasidenib, an inhibitor of isocitrate dehydrogenase 2 (FDA approved for R/R-AML with an IDH2 mutation on 1 August 2017), (iv) ivosidenib, an inhibitor of IDH1 (FDA approved for R/R-AML with an IDH1 mutation on 20 July 2018), (v) venetoclax, an inhibitor of BCL-2 that is independent of TP53 mutations (FDA approved for newly diagnosed AML in patients unfit for intensive chemotherapy in combination with HMAs or LDAC on 21 November 2018). Table 3 gives an overview of these substances and Figure 2 summarizes the mechanism of action of selected agents.

Below we will discuss all substances recently approved for the treatment of molecularly defined subgroups of AML, in addition to several other substances that are currently in phase II/III clinical trials that seem promising. Mutational analysis in most of the studies listed below was done via basic molecular testing (i.e., RT-PCR). However, since a broader use of NGS analysis is expected for routine clinical practice in the near future, we anticipate the use NGS based molecular profiling for the selection of targeted agents.

### 5.2. Targeting FLT3 Mutations

#### 5.2.1. Midostaurin (Novartis)

FMS-like tyrosine kinase 3 (FLT3; CD135) is a tyrosine kinase that regulates hematopoiesis. After binding of its ligand (fms-related tyrosine kinase 3 ligand) FLT3 is autophosphorylated and activates downstream pathways involved in proliferation, differentiation and apoptosis in hematopoietic stem cells [87]. FLT3 mutations (either tyrosine kinase domain (TKD) mutations or internal tandem duplications (ITD)) occur in about 25% of patients with AML and are associated with a worse prognosis [88,89,90,91]. Midostaurin is a kinase inhibitor of FLT3, but also binds other targets including c-KIT, PDGF-Rβ, VEGFR-2, and protein kinase C (Figure 2) [92]. Midostaurin has modest single agent activity in patients with AML with transient peripheral blast reduction in 42–72% of patients being the best response observed [61,93]. Consecutively, the drug was further evaluated in combination with cytotoxic agents.

Based on the promising results of a phase IB study of oral midostaurin (at a dose of 50 mg or 100 mg BID) combined with daunorubicin and AraC induction and consolidation in patients with newly diagnosed FLT3 mutated or wild type AML [63].

A large international randomized placebo controlled trial (CALGB 10603, RATIFY) was conducted [62]. In this trial, presence of an FLT3 (either TKD or ITD) mutations was a sine qua non for inclusion. Testing for FLT3 mutations was done according to the method described by Thiede et al. [89]: PCR was performed on genomic DNA with primers for the region of interest. The PCR products were then analyzed on a 3% agarose gel. Using the same PCR conditions with additionally labeled primers, the ratio of FLT3 ITD repetitions to wild type FLT3 was calculated and patients with a ratio >0.05 were termed positive. Patients aged 18–60 years were randomized to receive either standard induction therapy (daunorubicin 60 mg/m^2^ d1–3 + AraC 200 mg/m^2^ d1–7) in combination with oral midostaurin (50 mg BID on days 8–21) or placebo. After induction, patients in CR received four cylces of high dose AraC consolidation (3000 mg/m^2^ on day 1, 3 and 5, *q* = 28 d). Patients without CR received a second course of induction chemotherapy. Patients in CR after consilidation entered a maintenance phase with up to 12 cycles (*q* = 28 d) of midostaurin or placebo. Allogeneic transplantation was performed at the discreation of the investigator. The primary endpoint was overall survival. Of the 717 patients treated in the trial, median survival was 74.7 months in the midostaurin group and 25.6 months in the placebo group (HR 0.78, *p* = 0.009). Median event free survival was 8.2 and 3.0 months respectively (*p* = 0.002). Midostaurin added little toxicity to standard AML treatment, with anemia (93% vs. 88%, *p* = 0.03) and rash (14% vs. 8%, *p* = 0.008) being more common in the midostaurin group [62]. Based on these results, midostaurin was approved by the FDA on 28.04.2017 for the treatment of adult patients with newly diagnosed AML who are FLT3 mutation positive, as detected by an FDA approved test (LeukoStrat CDx FLT3 Mutation Assay (Invivoscribe Technologies Inc., San Diego, CA, USA); sensitivity >5% of total cells) in combination with standard AraC and daunorubicin induction and AraC consolidation-chemotherapy (https://www.fda.gov/drugs/informationondrugs/approveddrugs/ucm555756.htm; accessed on 4 January 2019). Marketing authorization for midostaurin was granted by the EMA on 18 September 2017 for FLT3 mutated AML using a validated test. (https://www.ema.europa.eu/en/medicines/human/EPAR/rydapt; accessed on 4 January 2019).

Of note, Zuffa et al. recently demonstrated that ultra deep sequencing (coverage of >8000 reads per base) has much higher sensitivity for detecting FLT3 mutations than conventional RT-PCR [94]. They used this approach in five patients with AML in whome no FLT3 mutation could be detected (i.e., FLT3 wild type) at diagnosis by RT-PCR, but relapsed with a FLT3 mutant clone (as detected by the same RT-PCR method). This prompted the authors to take a closer look and to reevaluate FLT3 status at initial diagnosis with a more sensitive method. Using ultra deep sequencing, a small FLT3 mutant clone (VAF = 0.2–2%) could be detected in all five patients in their diagnostic sample. Whether these patients might have profited from addition of an FLT3 inhibitor to their frontline treatment regimen can not be answered at the moment. However, this study creates a hypothesis that deep sequencing for therapeuticaly relevant mutations may have an impact on the choice of treatment and survival. Bearing this in mind, it would be desirable for updates of the current guidelines to include the minimum recommended coverage for NGS analyses. This would also help to make results from different groups more comparable.

#### 5.2.2. Gilteritinib (Astellas, Tokio, Japan)

Gilteritinib is a kinase inhibitor of FLT3 (Figure 2) with potent activity against FLT3 receptors with ITD and TKD mutations, which is also active in patients who harbor the resistance mutation FLT3-D835. Moreover, gilteritinib inhibits AXL, an oncogenic tyrosine kinase frequently overexpressed in AML that facilitates FLT3 activation and has been implicated in FLT3 inhibitor resistance [95,96]. After showing single agent activity in a phase 1/2 clinical trial in patients with R/R AML [68], results of the ADMIRAL trial (ClinicalTrials.gov Identifier NCT02421939), a randomized phase 3 trial of gilteritinib vs. investigator choice salvage chemotherapy have been recently reported. This trial included 138 adult patients with R/R AML having a FLT3 ITD, D835, or I836 mutation by the LeukoStrat CDx FLT3 mutation assay [97]. In the interim analysis of this trial, 21% of patients achieved a complete remission, or complete remission with partial hematologic recovery after a median follow-up of 4.6 months. The most ommon adverse events occurring in >20% of patients were myalgia, arthralgia, transaminase increase, fatigue, fever, non-infectious diarrhea, dyspnea, pneumonia, cough, edema, rash, nausea/vomiting, stomatitis, headache, hypotension, and dizziness. Based on the interim analysis of this trial FDA approved gilteritinib for the treatment of adult patients with R/R AML with a FLT3 mutation (as detected by an FDA-approved test) on 28 November 2018 (https://www.fda.gov/Drugs/InformationOnDrugs/ApprovedDrugs/ucm627045.htm; accessed on 4 January 2019). Gilteritinib is currently being tested in a randomized phase III trial (clinicaltrials.gov, identifier NCT02752035) alone versus in combination with azacitidine versus azacitidine alone in FLT3-mutant newly diagnosed AML. Recruitment commenced in June 2016 and efficacy results are expected in 2019.

#### 5.2.3. Quizartinib (Daiichi Sankyo, Tokio, Japan)

Quizartinib is an oral, highly potent and selective FLT3 inhibitor active against ITD mutations (Figure 2). In a Phase I study, the dose limiting toxicity was QTc time prolongation [67]. Two studies have been recently published presenting phase 2 data [64,65]. In the first study [64], efficacy and safety of quizartinib monotherapy was evaluated in two independent cohorts: cohort 1 consisted of patients at least 60 years of age with R/R AML within one year after first-line therapy, whereas cohort 2 consisted of patients who where at least 18 years of age with R/R AML after salvage chemotherapy (97%) or allogeneic stem cell transplantation (29%). Initial treatment of 17 patients with 200 mg/day yielded a higher rate of QTcF prolongation than expected (71%); therefore, lower doses (90 mg for women and 135 mg for men) were explored after a protocol amendment [64]. In cohort 1, 56% of FLT3 positive patients and 36% of FLT3 negative patients achieved composite complete remission (defined as CR + CRi). In cohort 2, 46% of FLT3 positive patients achieved composite complete remission, whereas 30% of FLT3 negative patients achieved composite complete remission. In cohort 2, quizartinib enabled 35% of 176 patients to bridge to haemopoietic stem cell transplantation. QTcF above 500 ms was reported in 17% and 15% of patients treated with 90 and 135 mg/day, respectively.

Although QTcF prolongation could be safely managed by pausing or discontinuing quizartinib in the above mentioned trial [64], different dosing schedules were explored [65]. In this phase 2b trial patients with R/R FLT3-ITD mutated AML were treated at 30 mg or 60 mg doses, respectively. Quizartinib monotherapy resulted in a CRc rate of 47% at both doses, which is similar to the above study. QTc time prolongation was less common (11% and 17%) with the lower doses. Overall survival (27 weeks) and duration of CRc (9.1 weeks) was comparable with higher doses [65].

Results of a phase 3 open label randomized trial (QANTUM-R; ClinicalTrials.gov Identifier: NCT02039726) testing single agent quizartinib in 367 patients with FLT3-ITD mutated R/R AML where presented at the 2018 ASH meeting [66]. Patients were randomized to either quizartinib 60 mg orally as monotherapy or one of three preselected investigators choice therapies (standard of care with either: low-dose AraC; mitoxantrone, etoposide, and intermediate-dose AraC; or fludarabine, AraC, and granulocyte-colony stimulating factor with idarubicin). Prior therapy with midostaurin was allowed, but prior treatment with all other FLT3 inhibitors was not. The hazard ratio of quizartinib relative to standard of care was significantly better (HR = 0.76; *p* = 0.0177). Median overall survival was 6.2 vs. 4.7 months for quizartinib vs. standard of care, respectively. The composite CR rate was 48% versus 27% (*p* = 0.0001) in the quizartinib and standard of care arms, respectively. Rates of treatment-emergent adverse events were comparable between the two arms. Encouragingly, patients older than 65 years profited slightly more from quizartinib compared to younger patients (HR 0.63 vs. 0.80; no p-value given) [66].

Based on the results of this trial, the FDA granted priority review for quizartinib for the treatment of adult patients with R/R FLT3-ITD AML in November 2018. The FDA is expected to decide on approval by May 25th, 2019. Quizartinib is currently also under expedited regulatory review with the Japan Ministry of Health, Labour and Welfare (MHLW) and the EMA for the same patient group (https://www.drugs.com/nda/quizartinib_181121.html; accessed on 4 January 2019).

A phase 3, double-blind, placebo-controlled, randomized study of quizartinib versus placebo in patients with newly diagnosed FLT3-ITD mutated AML administered in combination with induction- and consolidation chemotherapy and as maintenance therapy (up to 12 cycles) is ongoing (QuANTUM-First; ClinicalTrials.gov Identifier: NCT026686) and results are eagerly awaited.

Other FLT3 inhibitors include Lestaurtinib [69] and Crenolanib [71,72,73,74,75]. Clinical trials with crenolanib are ongoing and include a phase III randomized trial in newly diagnosed FLT3 mutated AML comparing standard induction + consolidation + either midostaurin or crenolanib (ClinicalTrials.gov Identifier: NCT03258931). Trial results are summarized in Table 3.

### 5.3. Targeting IDH (Mutations)

IDH mutations occur in about 20% of de novo AML. The prognostic effects of IDH mutations have been studied and appear to be mostly influenced by the location of the mutation (IDH1^R132^, IDH2^R140^, and IDH2^R172^) and the presence of other co-occurring mutations [98].

#### 5.3.1. Targeting IDH2 Mutations with Enasidenib (Celgene, Summit, NJ, USA)

Isocitrate dehydrogenase 2 (IDH2) is an enzyme which is part of the citric acid cycle, which reduces NADPH from NADP+ by catalyzing the oxidative decarboxylation of isocitrate to α-ketoglutarate (αKG) (Figure 2) [99]. IDH2 mutations (as well as TET2 loss of function mutations [100]) lead to the accumulation of the oncometabolite 2-hydroxyglutarate (2HG) (Figure 2). This causes epigenetic dysregulation leading to alterered gene expression and inhibition of cellular differentiation and maturation [101]. Mutations in IDH2 are found in approximately 12% of patients with AML (most commonly R140Q and R172K) [1]. Enasidenib binds to the mutated IDH2 protein leading to decreased levels of 2 HG and subsequent cellular differentiation [102].

In a phase I clinical trial (ClinicalTrials.gov, Identifier: NCT01915498) including 239 patients with R/R IDH2 mutated AML, enasidenib was tested at increasing doses (50–650 mg/day) [82]. The median age of the study population was 70 years (range 19–100). Grade 3 to 4 enasidenib-related adverse events included hyperbilirubinemia (12%) and IDH-inhibitor-associated differentiation syndrome (7%). Clinical signs of this syndrome are non-specific and include fever, edema, pleuropericardial effusions, respiratory symptoms and hypotension. IDH differentiation syndrome was manageable with steroids and withholding enasidenib until resolution [103]. The overall response rate (defined as CR + CRi + PR + morphologic leukemia free state) in this trial was 40.3%, with a median response duration of 5.8 months. Median overall survival was 9.3 months for the whole cohort and 19.7 months for patients who attained a CR (*n* = 34; 19.3%). IDH2 mutation type (i.e., R140 or R172) did not have any influence on treatment outcomes in this trial [82].

Based on the results of the above trial, FDA approved enasidenib on 1 August 2017 for the treatment of adult patients with R/R AML with an IDH2 mutation as detected by an FDA-approved test, at a starting dose of 100 mg once daily. FDA concurrently approved the RealTime IDH2 Assay (Abbot Molecular) as companion diagnostic method (https://www.fda.gov/Drugs/InformationOnDrugs/ApprovedDrugs/ucm569482.htm; accessed on 4 January 2019). EMA granted enasidenib orphan drug designation on 28 April 2016, and agreed to a pediatric investigation plan on 4 October 2017 (https://www.ema.europa.eu/en/medicines/human/orphan-designations/eu3161640; accessed on 4 January 2019).

Of importance, FDA issued a warning and health care alert on 29 November 2018, that signs and symptoms of the life-threatening side effect differentiation syndrome are not being recognized, despite this information being present in the prescribing information. FDA stressed the importance of early recognition and aggressive management of this side effect to lessen the likelihood of death (https://www.fda.gov/Drugs/DrugSafety/ucm626923.htm; accessed on 4 January 2019).

#### 5.3.2. Targeting IDH1 Mutations with Ivosidenib (Agios)

IDH1 mutations occur in about 6–10% of patients with AML [1]. In a recently published Phase I clinical trial (AG120-C-001, ClinicalTrials.gov Identifier NCT02074839), 179 patients with R/R AML with IDH1 mutations, confirmed using the Abbott RealTime IDH1 Assay, where treated with 500 mg of oral ivosidenib daily [83]. The most common adverse events related to ivosidenib treatment were prolongation of the QT interval (7.8%), IDH differentiation syndrome (3.9%), anemia (2.2%) and thrombocytopenia (3.4%). Complete remission or complete remission with partial hematologic recovery was achieved in 32.8% of patients with a median response duration of about 9 months. Among the 110 patients who were dependent on red blood cell and/or platelet transfusions at baseline, 41 (37.3%) became transfusion independent. Among the 34 patients who had a complete remission or complete remission with partial hematologic recovery, 7 achieved MRD negativity. No single gene mutation detected by NGS predicted clinical response or resistance to treatment [83].

Based on these results, ivosidenib received FDA approval on 20 July 2018, for the treatment of adult patients with R/R AML with a susceptible IDH mutation as detected by the Abbott RealTime IDH1 Assay, which was approved on the same day. (https://www.fda.gov/Drugs/InformationOnDrugs/ApprovedDrugs/ucm614128.htm; accessed on 4 January 2019). Ivosidenib received a positive aviso for orphan drug designation for the treatment of AML from EMA on 12 December 2016 (https://www.ema.europa.eu/en/search/search?search_api_views_fulltext=ivosidenib; accessed on 4 January 2019).

Taken together, IDH inhibition is becoming a standard of care in the treatment of AML. In most sequencing studies, IDH-1 and IDH-2 mutations were mutually exclusive. However, Platt et al. identified 21 out of 92 patients with AML, MDS or CMML with IDH mutations. Of these 21 patients, four had IDH1 and IDH2 mutations at the same time. This may have therapeutic implications, as the question arises if such patients should be treated with an IDH1 or an IDH2 inhibitor or the combination of both and further research is needed to answer these questions [104]. Mechanisms of resistance to IDH inhibition are a field of active research. Very recently, Harding et al. were able to show that some patients develop resistance to e.g., an IDH1 inhibitor by switching to the other isoform (i.e., switching to IDH2 in this example), indicating that repeated mutational may lead to strategies overcoming drug resistance [105].

### 5.4. Targeting TP53 (Mutations) with Drugs Whose Mechanism of Action Is TP53 Independent

#### 5.4.1. Overview of TP53 Mutations in AML

TP53 mutations in patients with AML occur in about 10% of the patients and are associated with 17p deletions as well as complex and monosomal karyotypes, which are all associated with an inferior prognosis, e.g., [106] Patients harboring TP53 mutations respond poorly to cytotoxic chemotherapy (response rate 28–42%), since TP53 activation is a critical step in the response to cytotoxic agents. Furthermore, relapses after achieving a response are common, especially in the elderly [107,108,109]. Therefore, drugs that kill AML cells independently of the presence of a TP53 mutation are of high interest in this difficult to treat patient population. The inhibitors discussed below are currently approved for all patients with AML regardless of their mutational profile. However, as we will show, certain genetically defined subgroups seem to profit especially from these inhibitors. Therefore, these drugs have been incorporated in this section.

#### 5.4.2. Targeting Bcl-2 with Venetoclax (AbbVie, Chicago, IL, USA)

B-cell leukemia/lymphoma-2 (Bcl-2), an anti-apoptotic protein commonly expressed in hematologic malignancies, has been shown to be involved in tumor survival and chemoresistance [110]. Venetoclax (ABT-199/GDC-0199) is a highly selective, orally bioavailable Bcl-2 inhibitor that has shown activity in bcl-2-dependent leukemia and lymphoma cell lines [111,112] Binding of venetoclax to Bcl-2 leads to the release of proapoptotic proteins (BIM, BAX), which then translocate to the mitochondria, ultimately leading to TP53 independent apoptosis (Figure 2) [113].

Venetoclax was first approved for the treatment of chronic lymphocytic leukemia after showing impressive results in relapsed and/or high risk patients [114]. In AML, Bcl-2 inhibition induces cell death of leukemic blasts and leukemia stem cells in vitro [115,116]. Based on the favourable results and approval of venetoclax in CLL, a phase II non randomized open94 label study of single agent venetoclax in patients with R/R-AML or de novo AML in patients unfit for intensive chemotherapy was conducted (ClinicalTrials.gov Identifier: NCT01994837) [85]. Thirty-two patients were treated with increasing doses of venetoclax (20 mg on week 1 day 1, 50 mg on day 2, 100 mg on day 3, 200 mg on day 4, 400 mg on day 5, and 800 mg on day 6 and daily thereafter). The overall response rate (defined as CR + CRi) was 19%; an additional 19% of patients demonstrated anti-leukemic activity not meeting IWG criteria (partial bone marrow response and incomplete hematologic recovery). Common G3/4 adverse events included nausea (6%), diarrhea (6%), febrile neutropenia (31%) and hypokalemia (22%). Patients with a sensitive bcl-2 index (defined as ≥35% of tumor cells expressing Bcl-2 and <40% of tumor cells expressing bcl-XL protein) had a longer time on study. Patients with an IDH Mutation (38% of patients) had a trend toward better response. This observation was in line with the previously reported Bcl-2 dependence in IDH mutated AML: Chan et al. demonstrated in vitro that the oncometabolite (R)-2-HG, which is produced by mutant IDH, inhibits the activity of cytochrome c oxidase in the mitochondrial electron transport chain thereby lowering the mitochondrial threshold to trigger apoptosis upon BCL-2 inhibition [117].

In order to predict possible in vivo response to venetoclax, Kontro et al. performed whole exome sequencing and gene expression profiling of bone marrow and peripheral blood samples obtained from patients with de novo or relapsed AML in comparison to bone marrow samples from healthy controls and CLL patients [118]. They discovered that responses to Bcl-2 inhibition seemed to correlate with mutations in chromatin modifiers as well as IDH and WT1. The same group also analysed gene expression and found that sensitivity to venetoclax was significantly correlated with overexpression of HOX A and B gene transcripts as patient samples with low expression of HOX A or B were generally resistant to venetoclax [118].

Very recently, durable responses with venetoclax in combination with azacitidine or decitabine in elderly (>65 years) patients with de novo AML have been reported in a phase IB study (www.clinicaltrials.gov, NCT02203773) [84]. Here, patients were randomized to receive either azacitidine (*n* = 29 in the 400 mg cohort and *n* = 37 in the 800 mg cohort) or dectiabine (*n* = 31 in the 400 mg cohort and *n* = 37 in the 800 mg cohort) as backbone. Venetoclax was coadministered daily with 20 mg/m^2^ of decitabine on days 1 to 5 or 75 mg/m^2^ of azacitidine on days 1 to 7, each 28-day cycle. Sixty, 74, and 11 patients received venetoclax at 400, 800, and 1.200 mg, respectively. The most common grade 3 or worse adverse events included febrile neutropenia (43%), thrombocytopenia (23%), and neutropenia (16%). At the 400 mg dose of venetoclax, the rate of complete remission/complete remission with incomplete blood cell count recovery was 73% (76% with azacitidine and 71% with decitabine). Even more encouraging, almost half of the responding patients achieved minimal residual disease negativity (assessed by a multiparameter flow cytometry) [84]. Furthermore, patients with poor risk karyotype and patients >75 years of age achieved similar CR rates compared to the total cohort (60% and 65%, respectively). Responses were also observed in patients carrying the TP53 mutation, with CR + CRi rates of 47%, median duration of CR + CRi of 5.6 months (95% CI, 1.2–9.4 months), and median OS of 7.2 months (95% CI, 3.7 months-NR). Median overall survival in the total cohort was 16.2 and 16.9 months for patients receiving the decitabine or azacitidine backbone, respectively; p-value not given). The estimated 6-month, 1-year, and 2-year OS rates for all patients were 80%, 59%, and 46%, which compares favourably to historical data with azacitidine alone [35,56,57,58,60].

For the sake of completeness we will briefly mention vosaroxin, which induces apoptosis of AML cells in a TP53-independent manner, highlighting its potential usefulness in TP53-mutated AML [119]. Despite interesting results from several clinical trials [120,121,122,123], the company developing the drug withdrew their marketing authorization application on 17 May 2017 after the initial documentation had been evaluated by EMA (https://www.genengnews.com/topics/drug-discovery/sunesis-withdraws-european-maa-for-aml-drug-vosaroxin/; https://www.esmo.org/Oncology-News/Withdrawal-of-the-Marketing-Authorisation-Application-for-Vosaroxin; accessed on 4 January 2019).

### 5.5. Targeting JAK2 Mutations

#### Ruxolitinib (Novartis, Basel, Switzerland)

Ruxolitinib is an inhibitor of Janus kinase 1 and 2 (JAK1 and 2) approved for the treatment of primary myelofibrosis and other chronic myeloid neoplasms [124,125] by the FDA and EMA. (https://www.ema.europa.eu/en/medicines/human/EPAR/jakavi; https://www.drugs.com/history/jakafi.html; accessed 6 January 2019). A phase I/II study tested the drug in 26 patients with R/R AML at increasing doses (50–200 mg BID) [126]. Infectious complications grade 3–4 were the most common toxicity observed in 58% of patients. Interestingly, one patient with seven prior therapies achieved a CR in this heavily pretreated cohort [126].

In another phase II study including 38 patients with R/R AML (www.clinicaltrials.gov; NCT00674479), ruxolitinib showed modest antileukemic activity with achievement of CR in 3 of 18 patients with post-myeloproliferative neoplasm AML [127].

So far ruxolitinib has shown modest response rates in AML. However, due to its good tolerability, further research in (JAK mutated) AML in combination with other substances is ongoing. Currently, four clinical trials evaluating ruxolitinib in AML are actively recruiting (clinicaltrials.gov; accessed 12 November 2018). Other JAK2 inhibitors currently in clinical trials include fedratinib (phase III), pacritinib, momelotinib and SB1518 (phase I and II). Allthough these inhibitors are mainly tested in chronic myeloid neoplasms, trials evaluating there use in (secundary) AML are underway.

## 6. Next Generation Sequencing for Response Assessment and Disease Monitoring

### 6.1. Minimal Residual Disease

The 2017 update of the ELN recommendations for assessing response in AML has introduced minimal residual disease (MRD) negativity as a new response category (CR without minimal residual disease) [24]. Two methods, by which MRD can be assessed, have been mentioned in this update: multicolor flow cytometry and RT-qPCR [128]. RT-qPCR quantitatively measures transcript frequency of a leukemia specific mutation or translocation detected at diagnosis compared to a “housekeeper” gene (usually ABL). MRD assessment by RT-qPCR has a high sensitivity with detection of 10^−4^ to 10^−6^ cells.

Complete remission requires the presence of < 5% myeloblasts in the bone marrow, absence of circulating blasts and hematologic recovery. Allmost all (94%) long term survivors of AML have achieved CR after primary treatment of AML (either with intensive chemotherapy or lower intensitiy treatment), indicating that CR is a prerequisite for long term cure [129]. However, many patients initially achieve a complete remission after induction chemotherapy (up to 80%) but eventually relapse, indicating that morphologic response assessment does not predict survival accurately [130]. Therefore, MRD negativity (assessed by flow cytometry or PCR) may be a more robust predictor of survival, as patients without detectable MRD have a lower risk of relapse [131,132,133,134]. Additionally, MRD positivity after achieving MRD negativity has a high predictive value for relapse (close to 100%) [135]. However, not every patient with AML has a target mutation detected by RT-PCR for MRD assessment and targeted resequencing does not account for changes in the genomic landscape (i.e., clonal evolution) [136,137,138,139].

As compared to RT-PCR, NGS panels have the advantage to cover a broader range of genes. In a recent large series including 482 patients with AML treated with intensive induction chemotherapy, Jongen-Lavrencic et al. performed targeted NGS with a 54 gene panel at diagnosis and after induction chemotherapy (in CR) [140]. They identified a mutation in 430 (89%) patients, which were randomized to a training (283 patients) and validation (147 patients) cohort. The most commonly mutated genes were NPM1, DNMT3A, FLT3 and NRAS. All patients had to be in morphologic CR after induction. Pretreatment mutations were still detectable in 51.4% of patients after induction chemotherapy. Mutations in DNMT3A, TET2 and ASXL1 were detected most often. Interestingly, persistence of these mutations did not affect survival at any threshold cutoff value, indicating that these mutations might represent nonleukemic clones (in context with the results discussed in the pathogenesis section of this review). On the other hand, detection of mutations in NRAS, PTPN11, KIT and KRAS were associated with a significantly higher relapse rate (55.4% vs. 31.9%; hazard ratio, 2.14; *p* < 0.001), as well as lower rates of relapse-free survival (36.6% vs. 58.1%; hazard ratio for relapse or death, 1.92; *p* < 0.001) and overall survival (41.9% vs. 66.1%; hazard ratio for death, 2.06; *P* < 0.001). Mutations in NRAS, PTPN11, KIT and KRAS were independtly associated with worse survival in multivariate analysis (hazard ratio for death, 1.64; *p* = 0.003).

A similar observation came from another study in standard risk AML patients with NPM1 mutations [141]. Here, 346 patients were analyzed using a 51 gene panel for targeted sequencing before and after completion of intensive induction chemotherapy. Persistence of NPM1-mutated transcripts in blood was present in 15% of the patients after the second chemotherapy cycle and was associated with a greater risk of relapse after 3 years (82% vs. 30%) and a lower rate of survival (24% vs. 75%). The presence of minimal residual disease was the only independent prognostic factor for death in multivariate analysis (hazard ratio, 4.84; 95% CI 2.57 to 9.15; *p* < 0.001). Interestingly, an increase in NPM1 transcripts upon sequential monitoring was a very reliable predictor of relapse (69 of 70 patients).

### 6.2. MRD Guided Treatment Modulation in Clinical Trials

As descriped above MRD negativity (assessed by flow cytometry or PCR) has been clearly associated with better clinical outcomes and its role in predicting survival is certainly increasing. MRD guided treatment approaches have been reported recently.

Zhu et al. prospectively evaluated an MRD-directed (using RT-PCR for RUNX1) consolidation approach for 116 patients with t(8;21) AML who were in first CR after induction chemotherapy [142]. In this study, patients who were MRD-positive after two courses of consolidation therapy were directed to allogeneic HSCT, whereas those who achieved MRD negativity were recommended to continue chemotherapy. The authors found that allogeneic HSCT improved outcomes for MRD-positive patients compared with MRD-positive patients who declined HSCT (cumulative incidence of relapse: 22.1% vs. 78.9%, *p* < 0.0001; disease-free survival: 61.7% vs. 19.6%, *p* = 0.001), whereas outcomes with chemotherapy were superior for MRD-negative patients (relapse rate 5.3%, disease free survival 94.7%) [142].

Similarly, Balsat et al. reported outcomes for ELN high-risk NPM1-mutant patients [143]. Here, patients who did not achieve MRD (defined as a 4 log reduction of NPM transcripts) after induction had a higher incidence of relapse (3 year cumulative incidence of relapse 65.8% vs. 20.5%, HR = 5.83, *p* < 0.001) and a shorter overall survival (HR = 10.99, *p* < 0.001). MRD positivity was an independent negative prognostic value in multivariate analysis (HR for survival 5.1, *p* < 0.001). Disease free survival and overall survival were significantly improved by alloHSCT in patients who were MRD positive (HR = 0.25, *p* = 0.047 for both variables), a benefit not observed in MRD negative patients (HR = 1.62 *p* = 0.419 and HR = 2.11 *p* = 0.26, respectively).

Very recently, results of the RELAZA2 trial, an open-label, multicentre, phase II trial have been reported [144]. In this study, patients with advanced MDS or AML, who had achieved a complete remission after conventional chemotherapy or allogeneic haemopoietic stem-cell transplantation, were prospectively screened for MRD during 24 months from baseline by either quantitative PCR for mutant NPM1, leukaemia-specific fusion genes (DEK-NUP214, RUNX1-RUNX1T1, CBFb-MYH11), or analysis of donor-chimaerism in flow cytometry-sorted CD34-positive cells. MRD-positive patients in confirmed complete remission received azacitidine. After six cycles, MRD status was reassessed and patients with major responses (MRD negativity) were eligible for a treatment de-escalation. Of 172 patients with AML 60 (30%) developed MRD during the 24-month screening period. 6 months after initiation of azacitidine, 31 (58%, 95% CI 44–72) of 53 patients were relapse-free and alive. Relapse-free survival at 12 months was 46% in the 53 patients who were MRD-positive and received azacitidine compared to 88% in the MRD negative population (HR = 6.6, *p* < 0.0001). The authors concluded that pre-emptive therapy with azacitidine can prevent or substantially delay haematological relapse in MRD-positive patients with MDS or AML [144].

These studies imply that a tailored postremission approach using MRD is feasible, but further information from prospective trials is needed. Also, the sensitivity of NGS based MRD assessment is expected to be improved with methods like droplet digital PCR in the future. As such, MRD assessment by NGS adds further information on relapse risk and survival. However, efforts to standardize MRD assessment in AML and guidelines on MRD based treatment modulation have to be developed in the future.

## 7. Conclusions

Next generation sequencing is an exciting tool that is significantly and continuously increasing our understanding of AML pathogenesis and treatment induced clonal evoluation during the course of the disease. Mutations detected by pretreatment molecular analysis can have major implications on prognostic stratification and treatment decisions, which already has led to changes in disease classification by WHO. Response assessment by MRD is superior to morphologic assessment and may help the clinician to guide decisions on treatment choice, initation time point, and/or intensifcation. For a broad use in daily clinical practice, standards for NGS based treatment decisions and monitoring have to be further defined in the future including but not limited to: (i) at what timepoints during the course of treatment should NGS analysis be performed? (ii) which target genes should be included in a NGS panel and which sequencing coverage should be used.

## Figures and Tables

**Figure 1 cancers-11-00252-f001:**
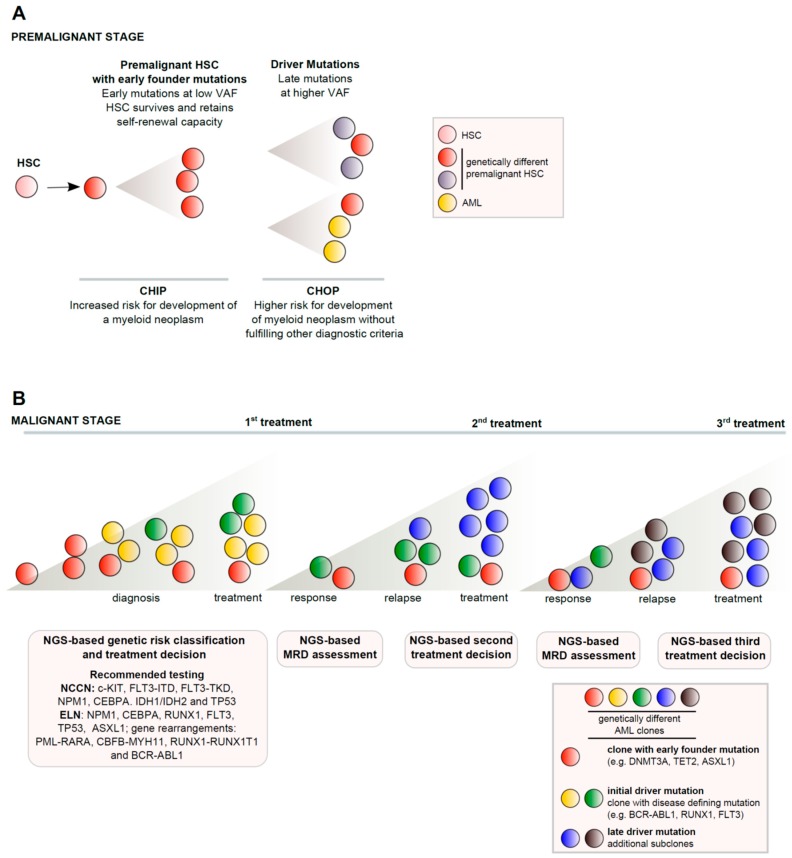
Pathogenesis of AML. (**A**) Premalignant stages preceeding to evolution of AML: early mutations in hematopoietic stem cells lead to clonal hematopoiesis (CHIP/CHOP) with genetically different premalignant stem cell subclones. (**B**) Subclonal genetic heterogeneity alongside AML development and progression is schematically depicted. NGS-based characterization of clonal and subclonal mutations is important for prognosis, treatment and response assessment (see text for explanations). HSC: Hematopoietic stem cell, VAF: Variant allelic frequency, CHIP: Clonal hematopoiesis of indeterminate potential, CHOP: Clonal hematopoiesis with substantial oncogenic potential, MRD: Minimal residual disease; NGS: Next generation sequencing, NCCN: National Comprehensive Cancer Network, ELN: European Leukemia Network.

**Figure 2 cancers-11-00252-f002:**
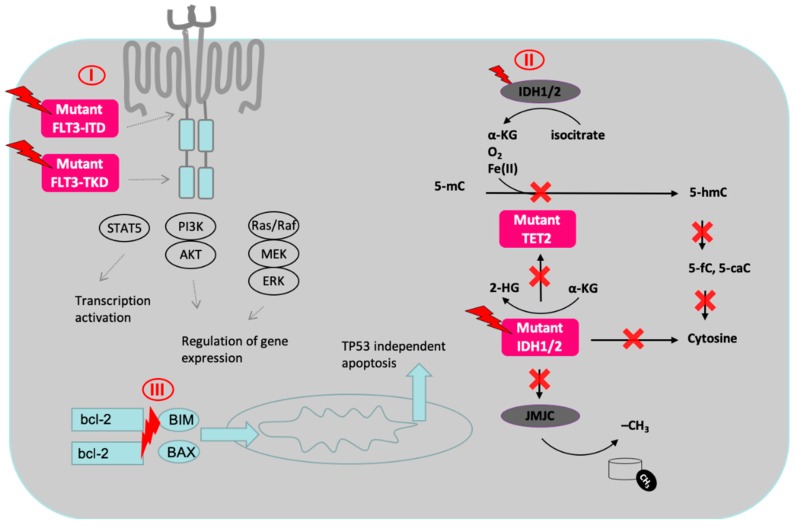
Cellular localization and mechanism of action of drugs targeting specific mutations. **I.** Targeting mutant FLT3 with TKI-inhibitors including midostaurin, quizartinib, crenolanib, gilteritinib and lestaurtinib. Mutations in FLT3 are present in approx. 30% of AML patients. **II.** Targting mutant IDH1 and IDH2 with IDH1 and IDH2 inhibitors including enasidenib and ivosidenib. Mutations in IDH are present in approx. in 30% of elderly AML patients. Both TET2 loss-of-function and IDH1/2 gain-of-function mutations result in reduced 5-hmC levels and in global promoter (and histone) hypermethylation. **III.** Targeting Bcl-2: Venetoclax binds to Bcl-2 thereby causing translocation of proapoptotic proteins (BIM, BAX) to the mitochondria. FLT3-ITD: fms like tyrosine kinase 3-internal tandem duplication; FLT3-TKD: fms like tyrosine kinase 3-tyrosine kinase domaine; STAT5: Signal transducer and activator of transcription 5; PI3K: Phosphoinositide 3-kinase; AKT: proteine kinase B; Ras/Raf: Rat sarcoma/rapidly accelerated fibrosarcoma; MEK: Mitogen-activated protein kinase kinase; ERK: extracellular signal–regulated kinases; bcl-2: B-cell lymphoma 2; BIM: Bcl-2-like protein 11; BAX: Bcl-2-associated X protein; IDH: isocitrate dehydrogenase; α-KG: alpha ketoglutarate; Fe(II): iron; 5-mC: 5-methylcytosine; 5-hmC: 5-hydroxymethylcytosine; TET2: Tet methylcytosine dioxygenase 2; 5-fC: 5-fluorcytosine, JMJC: Jumonji C -domain-containing proteins.

**Table 1 cancers-11-00252-t001:** Overview of commercially available NGS panels for AML with a list of included genes.

Qiagen Human Myeloid Neoplasms Panel ^1^	Illumina AmpliSeq Myeloid Panel ^2^	Quest Diagnostics LeukoVantage Panel ^3^	Oxford Gene Technology SureSeq myPanel NGS Custom AML ^1^
ASXL1 (full)	ASXL1 (full)	ASXL1	ASXL1 (full)
CEBPA (full)	CEBPA (full)	CEBPA	CEBPA (full)
DNMT3A (full)	DNMT3A (hotspot)	DNMT3A	DNMT3A (full)
FLT3 (full)	FLT3 (hotspot)	FLT3	FLT3 (full)
IDH1 (full)	IDH1 (hotspot)	IDH1	IDH1 (full)
IDH2 (full)	IDH2 (hotspot)	IDH2	IDH2 (full)
KIT (full)	KIT (hotspot)	KIT	KIT (full)
KMT2A (full)	KMT2A (fusion)	KMT2A	KMT2A (full)
KRAS (full)	KRAS (hotspot)	KRAS	KRAS (full)
NPM1 (full)	NPM1 (hotspot)	NPM1	NPM1 (full)
NRAS (full)	NRAS (hotspot)	NRAS	NRAS (full)
RUNX1 (full)	RUNX1 (full)	RUNX1	RUNX1 (full)
TET2 (full)	TET2 (full)	TET2	TET2 (full)
TP53 (full)	TP53 (full)	TP53	TP53 (full)
U2AF1 (full)	U2AF1 (hotspot)	U2AF1	U2AF1 (full)
WT1 (full)	WT1 (hotspot)	WT1	WT1 (full)
BCOR (full)	BCOR (full)	-	BCOR (full)
CALR (full)	CALR (full)	CALR	-
CBL (full)	CBL (hotspot)	CBL	-
CSF3R (full)	CSF3R (hotspot)	CSF3R	-
ETV6 (full)	ETV6 (full)	-	ETV6 (full)
EZH2 (full)	EZH2 (full)	EZH2	-
GATA1 (full)	-	GATA1	GATA1 (full)
JAK2 (full)	JAK2 (hotspot)	JAK2	-
MPL (full)	MPL (hotspot)	MPL	-
PHF6 (full)	PHF6 (full)	-	PHF6 (full)
PTPN11 (full)	PTPN11 (hotspot)	PTPN11	-
SETBP1 (full)	SETBP1 (hotspot)	SETBP1	-
SF3B1 (full)	SF3B1 (hotspot)	SF3B1	-
SRSF2 (full)	SRSF2 (hotspot)	SRSF2	-
ZRSR2 (full)	ZRSR2 (full)	ZRSR2	-
ABL1 (full)	ABL1 (hotspot)	-	-
BRAF (full)	BRAF (hotspot)	-	-
CREBBP (full)	CREBBP (fusion)	-	-
DDX41 (full)	-	DDX41	-
EGFR (full)	EGFR (fusion)	-	-
GATA2 (full)	GATA2 (hotspot)	-	-
HRAS (full)	HRAS (hotspot)	-	-
IKZF1 (full)	IKZF1(full)	-	-
KMD6A (full)	-	KMD6A	-
MYC (full)	MYC (expression)	-	-
MYD88 (full)	MYD88 (hotspot)	-	-
NF1 (full)	NF1 (full)	-	-
NTRK3 (full)	NTRK3 (fusion)	-	-
PDGFRA (full)	PDGFRA (fusion)	-	-
PRPF8 (full)	PRPF8 (full)	-	-
RB1 (full)	RB1 (full)	-	-
SH2B3 (full)	SH2B3 (full)	-	-
SMC1A (full)	SMC1A (expression)	-	-
STAG2 (full)	STAG2 (full)	-	-
91 genes available			

Full indicates all exons; hotspot indicates hotspot exons (not listed); fusion indicates RNA fusion partner (the repsective gene is not looked at on DNA level; not all of the fusion partners in the RNA panel listed); expression indicates the analyses of the mRNA expression level of these genes (the respective gene is not looked at on DNA level); ^1^ One can pick and choose genes from those listed. ^2^ Panel also includes an RNA-panel for fusion driver genes and expression genes (not fully listed). ^3^ The listed genes are derived from three panels (combination of the listed genes; no information about hotspot or full gene given on website).

**Table 2 cancers-11-00252-t002:** Summary of analysed genes in major NGS based sequencing studies in AML (*n* > 50 patients).

Author	Papaemanuil [1]	Tefferi [14]	Gangat [15]	Han Lin [16]	Hussaini [17]	Ruffalo [18]	Ley [2]	Lindsley [19]	Lindsley [19]	Lindsley [19]	Chun Ha [20]	Wang [21]	Welch [22]
**Year Published**	2016	2017	2018	2016	2018	2015	2013	2014	2014	2014	2016	2016	2016
**n pts**	1540	179	300	112	187	274	200	93 ^1^	no data ^2^	101 ^3^	60	95	54
**Median Age**	18–65	73	<70	43	no data	61,9	55	62	62	62	50	45	74
**n-Genes in Panel**	111	27	27	260	21	71	WGS/WES	82	82	82	54	410	264
**Gene**	**%pts**	**%pts**	**%pts**	**%pts**	**%pts**	**%pts**	**%pts**	**%pts**	**%pts**	**%pts**	**%pts**	**%pts**	**%pts**
TET2	13.3	25	26	10	15.3	14.0	8.0	20.0	9.0	14.0	8.0	9.5	14.8
ASXL1	4.61	30	27	16	20.7	5.0	2.5	32.0	3.0	no data	8.0	9.5	11.1
SMC1A	-	-	-	no data	-	-	no data	3.0	4.0	3.0	no data	no data	3.7
BCOR	2.34	-	-	no data	-	2.0	no data	8.0	2.0	1.0	no data	-	5.6
DNMT3A	24.9	10	13	15	14.8	21.0	26.0	19.0	28.0	27.0	8.0	16.8	14.8
IDH2	9.9	6	4	12	12	8.0	10.0	11.0	11.0	17.0	14.0	11.6	16.7
TP53-Others	7.21	13	12	no data	14.4	8.0	8.0	15.0	9.0	23.0	8.0	5.3	25.9
EZH2	3.12	4	3	no data	-	3.0	1.5	9.0	2.0	3.0	2.0	no data	1.9
KAT6A	-	-	-	no data	-	-	no data	-	-	-	-	-	0.0
IDH1	6.88	3	3	no data	10	8.0	9.5	11.0	11.0	17.0	6.0	4.2	9.3
JAK3	no data	-	-	no data	-	-	no data	-	-	-	no data	no data	0.0
KIT	4.61	2	no data	no data	10	2.0	4.0	3.0	no data	2.0	4.0	2.1	5.6
RUNX1	9.8	11	10	no data	15.2	9.0	10.0	31.0	11.0	11.0	-	5.3	16.7
SRSF2	6.04	16	13	no data	-	8.0	no data	20.0	1.0	10.0	no data	4.2	18.5
NF1	2.53	-	-	no data	-	3.0	no data	6.0	4.0	4.0	-	1.1	1.9
BCORL1	-	-	-	no data	-	-	no data	no data	no data	no data	no data	-	0.0
WT1	5.26	-	-	11	-	4.0	6.0	no data	no data	3.0	no data	11.6	7.4
FLT3 others	37.4	0.5	no data	21	11	16.0	28.0	19.0	28.0	16.0	32.0	18.9	5.6
NPM1	28.6	no data	no data	no data	11	16.0	27.0	5.0	30.0	16.0	24.0	21.5	11.1
IKZF1	no data	no data	no data	no data	-	-	no data	no data	no data	no data	2.0	1.1	0.0
KRAS	5.19	-	-	no data	-	4.0	12.0	8.0	4.0	11.0	no data	3.2	3.7
NRAS	19.0	no data	no data	no data	11.9	6.0	12.0	23.0	8.0	13.0	2.0	12.6	9.3
ATRX	0.39	-	-	no data	-	-	no data	no data	no data	no data	-	-	0.0
ZRSR2	0.78	no data	no data	no data	10	-	no data	8.0	no data	1.0	2.0	-	0.0
SF3B1	2.60	20	30	no data	10	5.0	no data	11.0	1.0	3.0	2.0	1.1	7.4
STAG2	4.48	-	-	no data	-	4.0	no data	14.0	2.0	6.0	4.0	5.3	5.6
U2AF1	2.47	16	no data	no data	10	6.0	no data	16.0	4.0	5.0	no data	7.4	9.3
SETBP1	-	3	3	no data	10	-	no data	5.0	no data	3.0	2.0	no data	3.7
PTPN11	8.51	no data	no data	no data	-	4.0	4.0	5.0	5.0	9.0	2.0	4.2	1.9
ABL1	-	-	-	no data	-	-	no data	-	-	-	-	no data	-
SMC3	-	-	-	no data	-	2.0	no data	2.0	4.0	2.0	4.0	2.1	1.9
JAK2	0.71	1	no data	no data	10	-	no data	no data	no data	no data	2.0	1.1	5.6
ETV6	1.43	-	-	no data	10	2.0	no data	no data	no data	no data	2.0	1.1	3.7
PRPF40B	no data	-	-	no data	-	-	no data	no data	no data	no data	-	no data	0.0
MLL	no data	-	-	no data	-	2.0	no data	no data	no data	no data	-	1.1	1.9
RAD21	3.70	-	-	no data	-	2.0	no data	2.0	3.0	4.0	4.0	3.2	1.9
GNAS	no data	-	-	no data	-	-	no data	no data	no data	no data	no data	-	0.0
CBL	2.73	1	3	no data	10	3.0	no data	5.0	2.0	4.0	2.0	no data	5.6
PHF6	3.05	-	-	no data	10	4.0	3.0	5.0	no data	1.0	no data	1.1	7.4
SUZ12	-	no data	no data	no data	-	-	no data	no data	no data	no data	-	-	0.0
CBLB	no data	-		no data	-	-	no data	no data	no data	no data	-	2.1	0.0
MPL	no data	no data	no data	no data	0	-	no data	no data	no data	no data	no data	no data	0.0
SF3A1	no data	-		no data	-	-	no data	no data	no data	no data	-	-	0.0
SH2B3	no data	no data	no data	no data	-	-	no data	no data	no data	no data	-	no data	0.0
U2AF2	0.13	-	14	no data	-	-	no data	no data	no data	no data	-	no data	0.0
DAXX	-	-	-	no data	-	-	no data	-	-	-	-	-	0.0
EED	-	-	-	no data	-	-	no data	no data	no data	no data	-	-	0.0
RB1	no data	-	-	no data	-	-	no data	-	-	-	-	no data	0.0
GATA1	no data	-	-	no data	-	-	no data	no data	no data	no data	no data	no data	0.0
SF1	no data	-	-	no data	-	-	no data	no data	no data	no data	no data	-	0.0
JAK1	-	-	-	no data	-	-	no data	-	-	-	-	no data	0.0
CBFB	-	-	-	no data	-	-	no data	-	-	-	-	no data	1.9
CEBPA	8.18	3	3	15	10	2.0	6.0	3.0	7.0	5.0	6.0	29.5	3.7

Hyphen indicates that the gene was analysed, but that the frequency in the analysed population was not given, respectively; ^1^ Secondary AML subgroup; ^2^ Treatment related AML gsubroup, ^3^ De novo AML subgroup.

**Table 3 cancers-11-00252-t003:** Summary of relevant clinical trials with targeted agents in AML.

Substance Group	Agent	Target	Ph	Patient Cohort	Schedule	ORR (%)	PFS (m)	OS (m)	A
**FLT 3 Inhibitors**	**Midostaurin** [61,62,63]	FLT3 (non mutated)	I	R/R or unfit	+ AZA	21	NR	6	Y
		FLT 3 mutated or WT	I	De novo	+ CTx	80	NR	NR	
		FLT3 mutated or WT	II	R/R or unfit	M	71 (mutated) 42 (WT)	NR	4.3	
		FLT3 mutated	III	De novo	+ CTx	59/53	26.7/15.5	74.7/25.6	
	**Quizartinib [64,65,66,67]**	FLT3 mutated	I	R/R	M	30	NR	3.5	Y
		FLT3 mutated or WT	II	R/R	M	47	2.2	6.7	
		FLT3 mutated or WT	II	R/R	M	74–77	3	6	
		FLT3 mutated	III	R/R	M vs. CTx	48 vs. 27	4 vs. 1.2	6.2 vs. 4.7	
	**Gilteritinib [68]**	FLT3 (mutated)	I	R/R	M	40	4.25	6.25	Y
	**Lestaurtinib** [69]	FLT3 (mutated)	III	De novo	+ CTx	97	40 vs. 36% (NS)	5 y OS 46 vs. 45% (NS)	N
	**Sunitinib** [70]	FLT 3 (mutated)	I/II	De novo elderly	+ CTx	59	12	18	N
	**Crenolanib** [71,72,73,74,75]	FLT ITD and D835	II	De novo	+ CTx	96	NR	Nre	N
			II	De novo	+ CTx	83	NR	Nre	
			II	R/R	+ CTx	67	NR	NR	
			II	R/R	+ CTx	36	NR	9.25	
			II	R/R	M	47	2	4.75	
	**Sorafenib** [76,77,78,79,80,81]	Multiple kinases	II	De novo	+ CTx	60 vs. 59	9 vs. 21	Nre	N
			I	R/R	M	10	NR	NR	
			I	After AlloTx in FLT3-ITD	M	NR	85% at 12 Mo	95% at 12 Mo	
			I	R/R after AlloTx with FLT3-ITD	+ AZA	50	NR	322 days	
			III	Maintenance After alloTx	M	NR	Nre vs. 30.9	NR	
**IDH Inhibitors**	**Enasidenib** [82]	IDH 2 (mutated)	I/II	R/R	M	40	6.4	9.3	Y
	**Ivosidenib** [83]	IDH 1 (mutated)	I	R/R	M	41	NR	NR	Y
**Bcl-2 Inhibitors**	**Venetoclax** [84,85]	Bcl-2	II	R/R or unfit	M	19	2.5	4.7	Y
			Ib	unfit	+ AZA or DAC	73	NR	18	
	**Obatoclax** [86]	Bcl-2 family	I/II	unfit	M	0	NR	NR	N

R/R: relapsed or refratory, unfit: patient unfit for intensive chemotherapy, PFS: progression free survival, OS: overall survival, NR: not reported, M: monotherapy, +CTx: in combination with high dose chemotherapy, A: approved, Y: yes, N: No, Nre: not reached, HDAC: histone deacetylase; AZA: azacytidine, FLT3: fms like tyrosine kinase 3, WT: wild type, AlloTx: allogeneic bone marrow transplantation, ITD: internal tandem duplication, DAC: decitabine; Substances are ordered by substance class in order of approval date. Within the subgroups, approved substances are listed first.

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
