# Peer review of "Next Generation Sequencing in AML—On the Way to Becoming a New Standard for Treatment Initiation and/or Modulation?"

_cancers, 2019, doi:10.3390/cancers11020252_

Round 1
Reviewer 1 Report
This review article by Leisch et al provides an update on the increasingly complex molecular classification of AML, presents a comprehensive overview of the molecularly targeted therapies that have been recently approved or are currently being investigated in clinical trials and discusses the important role that NGS may play in the near future for the diagnosis, risk stratification, treatment planning and response assessment in AML.
The manuscript is clear, comprehensive and well written. I only have minor comments:
1) in the overview of NGS systems, I would make it clear to the readers the difference between 'second generation sequencers' (Illumina and Thermo Fisher) and 'third generation sequencers (Pacific Bioscience and Oxford Nanopore) since only the first category is ready for diagnostic use. Pacbio and MinION/GridION etc instruments still have a high error rate.
2) Figure 2: please add abbreviations in the legend. The yellow font make numbers (I, II, III, IV) difficult to read. Also, the IVth category of drugs is not clearly represented.
3) Please provide the name of the company producing the FDA-approved IDH2 test
4) There are several typos throughout the manuscipt, in the Figures and in the Figure legends: line 139, 274, 344, 549, , 598, 722.. please read the text carefully to fix all the typos.
Author Response
Reviewer 1
Comments and Suggestions for Authors
This review article by Leisch et al provides an update on the increasingly complex molecular classification of AML, presents a comprehensive overview of the molecularly targeted therapies that have been recently approved or are currently being investigated in clinical trials and discusses the important role that NGS may play in the near future for the diagnosis, risk stratification, treatment planning and response assessment in AML.
The manuscript is clear, comprehensive and well written. I only have minor comments:
Comment 1: in the overview of NGS systems, I would make it clear to the readers the difference between 'second generation sequencers' (Illumina and Thermo Fisher) and 'third generation sequencers (Pacific Bioscience and Oxford Nanopore) since only the first category is ready for diagnostic use. Pacbio and MinION/GridION etc instruments still have a high error rate.
We have added this information to the text.
Comment to Figure 2: please add abbreviations in the legend. The yellow font make numbers (I, II, III, IV) difficult to read. Also, the IVth category of drugs is not clearly represented.
We have added the abbreviations in the legend and changed the font to red. Also, we removed the IV th category, since we only briefly mention this pathway throughout the text.
Comment 3: Please provide the name of the company producing the FDA-approved IDH2 test
We have added this information to the text.
Comment 4: There are several typos throughout the manuscipt, in the Figures and in the Figure legends: line 139, 274, 344, 549, , 598, 722.. please read the text carefully to fix all the typos.
We corrected the typos as indicated.
Reviewer 2 Report
In their manuscript entitled "Next generation sequencing in AML-On the way to becoming a new standard for treatment initiation and/or modulation ?", Leish et al propose a very interesting review of the litterature regarding the interest of molecular analysis to improve clinal care of AML.
The paper is well written and will be useful for clinicians and also molecular biologists. However, I would like to propose some potential improvements :
1/ The chapter 4 entitled "Next generation sequencing to guide treatment in AML?" focus mainly on the description of early clinical trials of next therapeutic agents in AML. I would suggest to reduce this part of the review which is a little bit out of scope : indeed, most of the molecular-based strategies do not rely on NGS, but instead on basic molecular testing (for FLT3 ITD, IDH...). Moreover, BCL2 inhibitors can not be considered a NGS guided treatment so far, and should be associated with other non molecularly targeted drugs (such as gemtuzumab and so on). For this chapter, it is required to precise the name of the pharma for each drug, given the conflict of interests declared by the authors
2/ The Nature Genetics paper by Gerstung et al is cited, but should be more extensively described. It is a real change of paradigm in prognostic assessment of AML, based on an innovative integration of diagnostic parameters, enabling personalized assesment of prognosis. Please cite also Huet et al (Blood 2018) doi: https://doi.org/10.1182/blood-2018-03-840348 , which validates the Gerstung paper in the real life setting.
3/ line 38 : "on advances in molecular sequencing techniques " : molecular should be removed
4/ line 86 : "may range anywhere from 2h to two days " to a few days seems more realistic (at least in my lab !)
5/ Table 1 : could you precise if the genes are entirely sequenced or only in mutational hotspot ?
6/ Table 2 : could you remove the data from the unpublished reference (especially because the frequences of mutations are very different from other studies)
7/ regarding the CHIP : please cite the NEJM paper showing increased cardiovascular events in CHIP patients. I'm not sure that the CHOP term is really convincing...
8/line 207 : "CEBPA mutational status (assessed by RT-PCR) " : i'm surprised by this sentence, because as far as I know it is not possible to use RT-PCR to assess CEBPA mutationnel status... please precise or correct
9/ regarding the Welch paper (line 271 and following), the absolute number of patients should be precised (not very high if I remember well)
10/ line 377-378 : "and apoptosis in hematopoietic bone marrow stem cells " : bone marrow should be removed
11/ regarding the companion tests approved by FDA : is it the same policy with EMA ? Maybe a critical word about this commercial strategy, which might threaten the role of academic biology labs (if you think the same, it is just a suggestion)
12/ line 419 and following : is it really helpful to treat a very small clone ? open question that you could discuss ?
13/ line 515 : the sentence is wrong : tet 2 mutations do not produce 2HG
14/ line 540 : there is a recent JAMA Oncol paper regarding the differenciation syndrome under IDH inhibitors (fathi et al, 2018)
15/ line 566 : cite also (and discuss) the cancer discovery paper http://cancerdiscovery.aacrjournals.org/content/8/12/1540
16/ results of venetoclax in CLL are out of scope of this review
17/line 667 : cite the Gerstung paper to decide the allograft indication
18/ regarding MRD : nearly all the patients can be followed by FACS... so the use of NGS is less needed. Maybe a word about the optimal timing of MRD assessment ?
19/ line 712-717 : NPM1 MRD is assessed by RTqPCR, not NGS. So this is not really in the scope of the review... also true for line 733 and following, and for CBF AML (line 725 and following)
20/ maybe discuss the low sensitivity of NGS for MRD assessment ?
Author Response
Reviewer 2:
Comments and Suggestions for Authors
In their manuscript entitled "Next generation sequencing in AML-On the way to becoming a new standard for treatment initiation and/or modulation ?", Leish et al propose a very interesting review of the litterature regarding the interest of molecular analysis to improve clinal care of AML.
The paper is well written and will be useful for clinicians and also molecular biologists. However, I would like to propose some potential improvements :
1/ The chapter 4 entitled "Next generation sequencing to guide treatment in AML?" focus mainly on the description of early clinical trials of next therapeutic agents in AML. I would suggest to reduce this part of the review which is a little bit out of scope : indeed, most of the molecular-based strategies do not rely on NGS, but instead on basic molecular testing (for FLT3 ITD, IDH...). Moreover, BCL2 inhibitors can not be considered a NGS guided treatment so far, and should be associated with other non molecularly targeted drugs (such as gemtuzumab and so on). For this chapter, it is required to precise the name of the pharma for each drug, given the conflict of interests declared by the authors
We agree with the reviewer, that most of the targeted agents do not rely on the result of NGS testing at the moment. However, given the current trend of increasing use of NGS, we anticipate that mutational profiling of AML by NGS will be the bases for the selection of targeted agents in near future.
We also agree, that the treatment part of this review is very detailed. To us as being practicing physicians, this seemed to be the most relevant information we wanted to share with our colleagues. We tried to shorten this part were possible.
We completely understand the reviewers point of view regarding bcl-2 inhibitors. In our understanding, bcl-2 inhibitors are targeted agents that show particular efficacy in certain genetically defined subgroups. Therefore, we included bcl-2 inhibitors in the section for NGS guided treatments. We added a point of discussion to the text to make this point clearer.
The name of the producing pharmaceutical company was added to the text.
2/ The Nature Genetics paper by Gerstung et al is cited, but should be more extensively described. It is a real change of paradigm in prognostic assessment of AML, based on an innovative integration of diagnostic parameters, enabling personalized assesment of prognosis. Please cite also Huet et al (Blood 2018) doi: https://doi.org/10.1182/blood-2018-03-840348 , which validates the Gerstung paper in the real life setting.
We agree with the reviewer and explained this paper in more detail and added the citation requested.
3/ line 38 : "on advances in molecular sequencing techniques " : molecular should be removed
We made the suggested corrections.
4/ line 86 : "may range anywhere from 2h to two days " to a few days seems more realistic (at least in my lab !)
We made the suggested corrections.
5/ Table 1 : could you precise if the genes are entirely sequenced or only in mutational hotspot ?
We added this information to the table legend
6/ Table 2 : could you remove the data from the unpublished reference (especially because the frequences of mutations are very different from other studies)
We removed the unpublished data
7/ regarding the CHIP : please cite the NEJM paper showing increased cardiovascular events in CHIP patients. I'm not sure that the CHOP term is really convincing...
We added this citation. In our opinion, there is a relevant clinical difference implicated in the terms CHIP and CHOP. However, since these terms are fairly new, there is currently room for interpretation used by different authors. We added this information to the text.
8/line 207 : "CEBPA mutational status (assessed by RT-PCR) " : i'm surprised by this sentence, because as far as I know it is not possible to use RT-PCR to assess CEBPA mutationnel status... please precise or correct
We changed this aspect in the text
9/ regarding the Welch paper (line 271 and following), the absolute number of patients should be precised (not very high if I remember well)
We added the number of patients to the text
10/ line 377-378 : "and apoptosis in hematopoietic bone marrow stem cells " : bone marrow should be removed
We made the suggested corrections.
11/ regarding the companion tests approved by FDA : is it the same policy with EMA ? Maybe a critical word about this commercial strategy, which might threaten the role of academic biology labs (if you think the same, it is just a suggestion)
We added information about the EMA policy.
12/ line 419 and following : is it really helpful to treat a very small clone ? open question that you could discuss ?
We do not know the answer to this question as you most likely anticipated, but the results leave room for speculation. We added a discussion to the text.
13/ line 515 : the sentence is wrong : tet 2 mutations do not produce 2HG
We corrected the sentence
14/ line 540 : there is a recent JAMA Oncol paper regarding the differenciation syndrome under IDH inhibitors (fathi et al, 2018)
This paper is already cited in the text.
15/ line 566 : cite also (and discuss) the cancer discovery paper http://cancerdiscovery.aacrjournals.org/content/8/12/1540
We thank the reviewer for this very relevant input and added the information and the citation to the text.
16/ results of venetoclax in CLL are out of scope of this review
We removed this section from the review.
17/line 667 : cite the Gerstung paper to decide the allograft indication
We added this paper
18/ regarding MRD : nearly all the patients can be followed by FACS... so the use of NGS is less needed. Maybe a word about the optimal timing of MRD assessment ?
We agree with the reviewer that FACS analysis is powerful tool for MRD assessment, but in our opinion out of scope of this review. Regarding the timinig of MRD assessment, we added some points of discussion to the text. Guidelines regarding this question are avaited.
19/ line 712-717 : NPM1 MRD is assessed by RTqPCR, not NGS. So this is not really in the scope of the review... also true for line 733 and following, and for CBF AML (line 725 and following)
We agree with the reviewer that MRD asessment in many trials was done with RT-PCR. However, the cited studies provide the reader with a proof of principle that PCR based methods can be used for the measurement of MRD. The discussed results might also be seen using NGS based approaches as shown by the study by Jongen-Lavrencic et al.
20/ maybe discuss the low sensitivity of NGS for MRD assessment ?
We added a point of discussion regarding this Topic.